# Computation-based regulation of excitonic effects in donor-acceptor covalent organic frameworks for enhanced photocatalysis

Yunyang Qian[1,3], Yulan Han[2,3], Xiyuan Zhang[1], Ge Yang[1], Guozhen Zhang [2] & Hai-Long Jiang [1] ✉

The strong excitonic effects widely exist in polymer-semiconductors and the large exciton binding energy ($E_b$) seriously limits their photocatalysis. Herein, density functional theory (DFT) calculations are conducted to assess band alignment and charge transfer feature of potential donor-acceptor (D-A) covalent organic frameworks (COFs), using 1,3,5-tris(4-aminophenyl)triazine (TAPT) or 1,3,5-tris(4-aminophenyl)benzene (TAPB) as acceptors and terephthaldehydes functionalized diverse groups as donors. Given the discernable D-A interaction strengths in the D-A pairs, their $E_b$ can be systematically regulated with minimum $E_b$ in TAPT-OMe. Guided by these results, the corresponding D-A COFs are synthesized, where TAPT-OMe-COF possesses the best activity in photocatalytic $H_2$ production and the activity trend of other COFs is associated with that of calculated $E_b$ for the D-A pairs. In addition, further alkyne cycloaddition for the imine linkage in the COFs greatly improves the stability and the resulting TAPT-OMe-alkyne-COF with a substantially smaller $E_b$ exhibits ~20 times higher activity than the parent COF.

Photocatalysis converting solar energy into chemical energy has been recognized to be one of the most important solutions toward the present energy and environmental challenges[1,2], where photocatalytic hydrogen production via solar water splitting plays a key role. Currently, traditional inorganic semiconductor photocatalysts for hydrogen production, such as metal oxides, nitrides, phosphides, and sulfides, are the main focus of research[3,4]. In recent years, polymer semiconductor photocatalysts have drawn great attention by virtue of their low toxicity and cost[5–7]. They are polymerized by small molecular monomers via covalent bonds. Accordingly, the delocalization of π electrons caused by conjugation enables most polymers good visible light response, which is highly desired in photocatalysis[5,7]. However, the low dielectric properties of polymers often lead to strong excitonic effects (the Coulomb interaction between photoinduced electrons and holes), which are detrimental to photocatalysis[7–10]. Exciton binding energy ($E_b$), being an important parameter to evaluate excitonic effects

of a photocatalyst, reflects the strength of Coulomb interaction between electrons and holes[11]. Therefore, the rational fabrication of polymer semiconductors with reduced exciton binding energy would be of great importance to achieve efficient photocatalysis.

To reduce the $E_b$ in polymer semiconductors, quite a few strategies such as constructing heterojunctions[12], homojunctions and donor-acceptor (D-A) molecular junctions have been developed[8,13–15]. They can accelerate the dissociation of excitons and increase the photocatalytic efficiency. In the field of organic solar cells, D-A molecular junctions have received intensive attention due to their ease of construction and structural diversity, and have been established as effective methods in reducing the $E_b$ to optimize the photovoltaic process from theoretical works[16–18]. In photocatalysis, the photocatalyst constructed by the D-A junctions is excited by light, causing the electrons from the donor to migrate to the electrophilic acceptor, giving rise to intermolecular charge transfer, which thereby inhibits

[1]Department of Chemistry, University of Science and Technology of China, Hefei, Anhui 230026, P. R. China. [2]Department of Chemical Physics, University of Science and Technology of China, Hefei, Anhui 230026, P. R. China. [3]These authors contributed equally: Yunyang Qian, Yulan Han. ✉e-mail: jianglab@ustc.edu.cn

the recombination of the carriers in the polymer caused by the strong excitonic effects[13–15].

In fact, the construction of D-A junction has been well developed in conjugated microporous polymers (CMPs) and linear organic polymers[5,13–15]. Unfortunately, they are disordered structures and usually lack long-range ordered nature, which make this molecular-based precise regulation very challenging. In this context, covalent organic frameworks (COFs), as a type of crystalline polymers with precise and tailorable structures[19–30], possess significant advantages to regulate the $E_b$ through D-A strategy and the corresponding activity toward photocatalysis. Theoretically, although the abundant D-A-type COFs (simply as D-A COFs) can be synthesized with an infinite selection of organic monomers[31–41], as a matter of fact, they might possess distinctly different photocatalytic activity due to their discriminative D-A interactions and $E_b$. Therefore, it is necessary to find the optimized COF with $E_b$ as low as possible from the numerous D-A structures. However, given the current challenges in the synthesis of COFs with high crystallinity[42–45], to date, there has yet been a report on the synthesis of a series of D-A COFs, and the systematic investigation of their relationship between D-A interaction and $E_b$, as well as their influences on the resulting photocatalysis. In this context, theoretical prediction might be an effective strategy to screen out the targeted D-A structure prior to the COF synthesis[46,47], which would be rational, time- and energy-saving to achieve the D-A COFs with optimized energy band structure and exciton binding energy for enhanced photocatalysis.

Herein, two amine-based monomers (1,3,5-tris(4-aminophenyl) triazine, simply as TAPT and 1,3,5-tris(4-aminophenyl)benzene, simply as TAPB) and five tereph-thaldehydes bundled with different groups (-Cl, -H, -OCCH, -OH, and -OMe) have been adopted to construct the corresponding COFs by DFT simulations (Fig. 1). It is found that TAPT monomer can be integrated with all aldehyde-based monomers to afford the D-A structures, while TAPB can give the D-A structure with the aldehydes with -OH and -OMe groups only. By comparing the differential charges at the ground and excited states, the charge separation degree of the D-A pairs (TAPT-OMe, TAPT-OH, TAPT-OCCH, TAPT-H, TAPT-Cl, and TAPB-OMe) gradually decreases, while the calculated $E_b$ follows an increasing trend, both of which support that photocatalytic activity will be in an order of TAPT-OMe > TAPT-OH > TAPT-OCCH > TAPT-H > TAPT-Cl > TAPB-OMe. To verify this prediction, four representative COFs, namely TAPT-OMe-COF, TAPT-H-COF, TAPT-Cl-COF and TAPB-OMe-COF, have been synthesized for

photocatalytic H$_2$ production. Experimental results are in good agreement with the prediction and TAPT-OMe-COF exhibits the highest activity with a H$_2$ production rate of 450 μmol g$^{-1}$h$^{-1}$. Although these imine COFs are not sufficiently stable in the photocatalysis, the imine bond can be reinforced to more stable quinoline linkage by the aza-Diels-Alder cycloaddition reaction with phenylacetylene. The activity of resulting TAPT-OMe-alkyne-COF sharply reaches 7875 μmol g$^{-1}$h$^{-1}$, ~18 times higher activity than TAPT-OMe-COF, and presents excellent photocatalytic stability and recyclability. Further DFT calculations show that, in contrast to the parent imine COFs, charge separation is further improved and the exciton binding energy is significantly reduced in the alkyne-modified COFs from the temperature-dependent photoluminescence spectra and calculation, accounting for their enhanced activity. To our knowledge, this is the first report on computation-based regulation of excitonic effects in D-A COFs for enhanced photocatalysis.

## Results
### Construction of D-A pairs
Imine bond, one of the most common linkages in COFs, is adopted to construct the COF platform[48–50]. The classical amine monomers, TAPB and TAPT, have been chosen as amine-based modules[33]. Five functionalized terephthalaldehydes with C$_2$ symmetry, including 2,5-dimethoxyterephthalaldehyde (OMe), 2,5-dihydroxyterephthalaldehyde (OH), 2,5-dichloroterephthalaldehyde (Cl), 2,5-diethynyloxyterephthalaldehyde (OCCH) and terephthalaldehyde (H), are chosen as the other type of modules. Accordingly, a total of ten possible combinations are evaluated for their possible construction of D-A COFs. As a matter of fact, to form D-A COFs, amine and aldehyde-based modules must meet a special requirement for frontier orbital alignment that, the highest occupied molecular orbital (HOMO) and the lowest unoccupied molecular orbital (LUMO) levels of the donor module are required to form the staggered feature with the HOMO and LUMO levels of the acceptor module (Supplementary Fig. 1)[47,51]. In this way, the LUMO and HOMO of D-A COFs are primarily contributed by the acceptor and the donor, respectively. In comparison, the HOMO and LUMO of the two modules presenting the straddling type would lead to the formation of D-D type COF, of which the HOMO and LUMO are both mainly contributed by the same module. Apparently, the D-A COFs are beneficial to photoinduced intermolecular charge transfer and therefore an ideal design for enhanced photocatalysis.

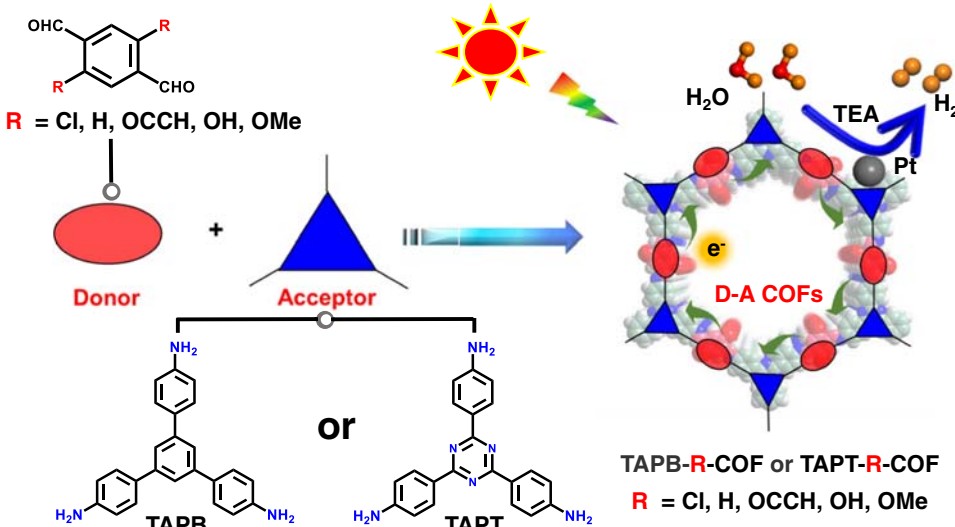

**Fig. 1 | Schematic illustration.** Construction of D-A COFs composed by amine monomers and functionalized tereph-thalaldehydes for photocatalytic H$_2$ production. The TAPB, TAPT and TEA are abbreviations for 1,3,5-tris(4-aminophenyl)benzene, 1,3,5-tris(4-aminophenyl)triazine and triethylamine, respectively.

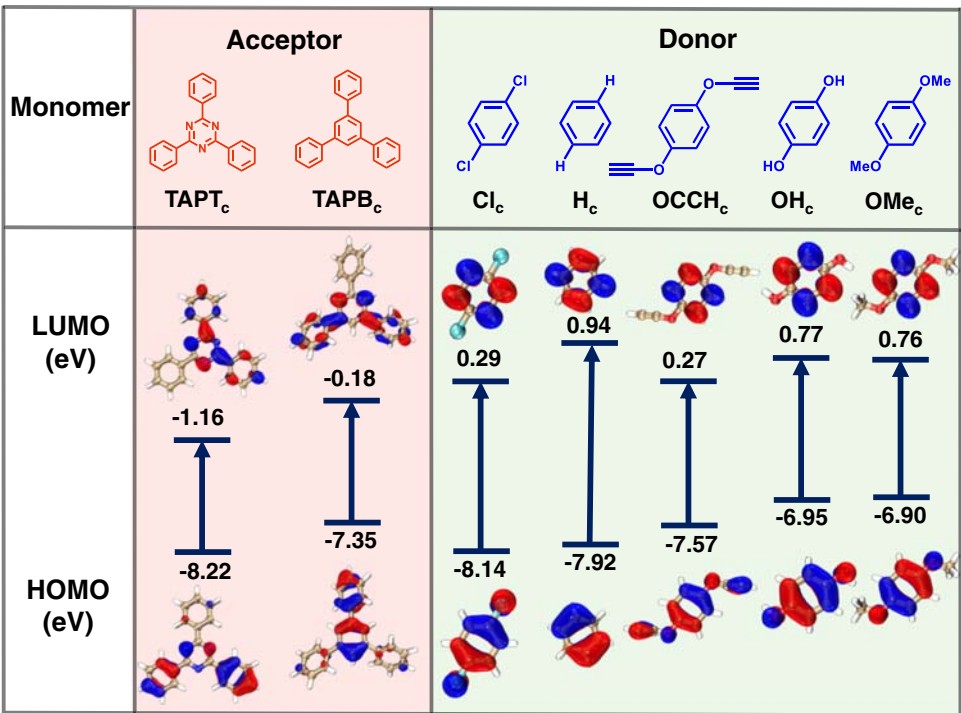

**Fig. 2 | HOMO and LUMO energy level diagram for the two types of building blocks for constructing D-A pairs.** The TAPT$_c$ and TAPB$_c$ represent the core part (without the amine group) of the amino-based monomers (TAPT and TAPB), and the Cl$_c$, H$_c$, OCCH$_c$, OH$_c$ and OMe$_c$ represent the core part (without the aldehyde group) of the terephthalaldehyde-based monomers.

To find out which amine-aldehyde pairs can form effective D-A pairs, we calculate the HOMO and LUMO energy levels of all the monomers of interest (Fig. 2; Supplementary Table 1 and 2). Both HOMO and LUMO of 2,4,6-triphenyl-1,3,5-triazine (TAPT$_c$, the core part of TAPT) are lower than the corresponding orbitals of all aldehyde-based modules, therefore, a staggered structure for D-A pairs can be constructed. In contrast, the 1,3,5-triphenylbenzene (TAPB$_c$, the core part of TAPB) module can provide D-A pairs with OH- or OMe-functionalized module only, as the LUMO of TAPB$_c$ is lower than that whereas the HOMO is higher than that of the modules with -Cl, -H and -OCCH groups. Along with the above energy level calculation of the monomers, the six D-A pairs, namely TAPT-Cl, TAPT-H, TAPT-OCCH, TAPT-OH, TAPT-OMe and TAPB-OMe (the energy level of OH$_c$ is similar to the OMe$_c$ (Fig. 2); therefore, TAPB-OMe only as a representative has been further investigated), are chosen for further calculation.

## Strength of D-A interaction

The six D-A pairs are chosen to investigate the possible electronic excitation transition, including charge transfer and local excitation modes (Supplementary Fig. 2)[52]. The charge transfer mode would reflect the efficient separation of electrons and holes and favorable photocatalytic performance. Based on the hole-electron theory, these two modes can be identified by S/D value[52,53], where S represents calculated overlap integral of hole-electron distribution, and D indicates the calculated distance between the hole and electron centroids (Supplementary Table 3). Therefore, the smaller S paired with the larger D means the more evident charge transfer. The computed S/D values follow the increasing trend of TAPT-OMe (0.36), TAPT-OH (0.42), TAPT-OCCH (0.61), TAPT-H (0.95), TAPT-Cl (1.09) and TAPB-OMe (2.10), suggesting that TAPT-OMe is the most favorable one for charge transfer.

Charge density difference between the ground and the excited states are adopted to visualize the electron transfer within D-A pairs

(Fig. 3; For each COF: $\Delta\rho$ is shown at the left side; centroids of charges are shown at the right side)[52,54]. From a more intuitive representation of the charge density difference (centroids of charges), the positive and negative centroids for D-A pairs are located on the aldehyde- and amine-based segments, respectively manifesting their electron donor and acceptor roles, which matches the above energy level calculation results (Fig. 2). To look into the subtle changes in this system, the distance of charge transfer ($D_{CT}$) based on the *electron density variation* and the amount of transferred electron during electron excitation have been calculated (Fig. 3). The larger $D_{CT}$ value and the higher amount of transferred electron denote the stronger D-A pairs. Among these pairs, TAPT-OMe features the largest $D_{CT}$ value (3.33 Å) and the highest amount of transferred electron (0.28 e). The integrated $D_{CT}$ and electron transfer amount, as the descriptors for D-A interaction, exhibit a descending order as: TAPT-OMe > TAPT-OH > TAPT-OCCH > TAPT-H > TAPT-Cl > TAPB-OMe, which is similar to the trend of the S/D values. Therefore, TAPT-OMe possesses the strongest D-A interaction, inferring the best photocatalytic performance of the corresponding COF.

## Exciton binding energy of D-A pairs

Exciton binding energy, which directly relates to excitonic effects of these COFs, has been also investigated by DFT calculations. The $E_b$ is calculated by the formula: $E_b = E_{fund} - E_{opt}$, where $E_{fund}$ is the fundamental gap and defined as the difference between ionization potential (IP) and electron affinity (EA), and $E_{opt}$ is the optical gap and defined as the difference between the ground state and the lowest excited state (Supplementary Fig. 3)[55]. For $E_{fund}$, IP mainly depends on the nature of the donor: the donor that is more likely to lose an electron has a lower level of IP; EA is mainly dominated by the acceptor: the acceptor that binds an extra electron stronger has a higher level of EA[47]. For $E_{opt}$, stronger D-A interaction tends to generate a smaller gap while weaker D-A interaction would lead to a larger gap (Supplementary Table 4 and 5)[47]. Intriguingly, $E_{fund}$ and $E_{opt}$ share similar trends among all D-A pairs of concern. The relative value of $E_b$ is correlated with relative strength

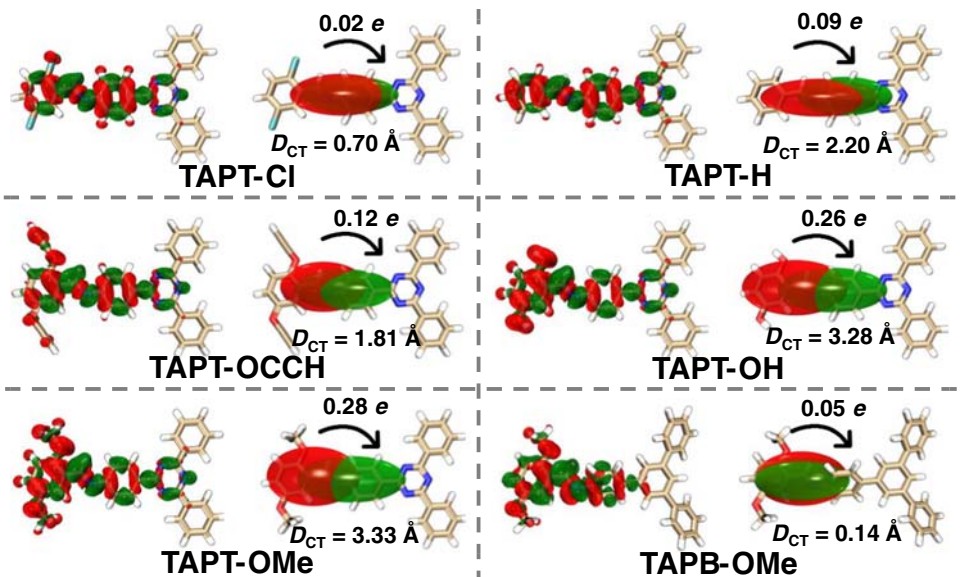

**Fig. 3 | Computed charge density difference between the ground and the excited states of the D-A pairs at an isovalue of 0.0005 a.u.** (For each COF: Δρ is shown at the left side; centroids of charges are shown at the right side). The green and red represent increase and decrease in electron density, respectively.

Quantitative charge-transfer analysis is based on the atomic dipole corrected Hirshfeld (ADCH) atomic charges. $D_{CT}$ represents the distance between the barycenter of the density increment and depletion regions upon electronic excitation.

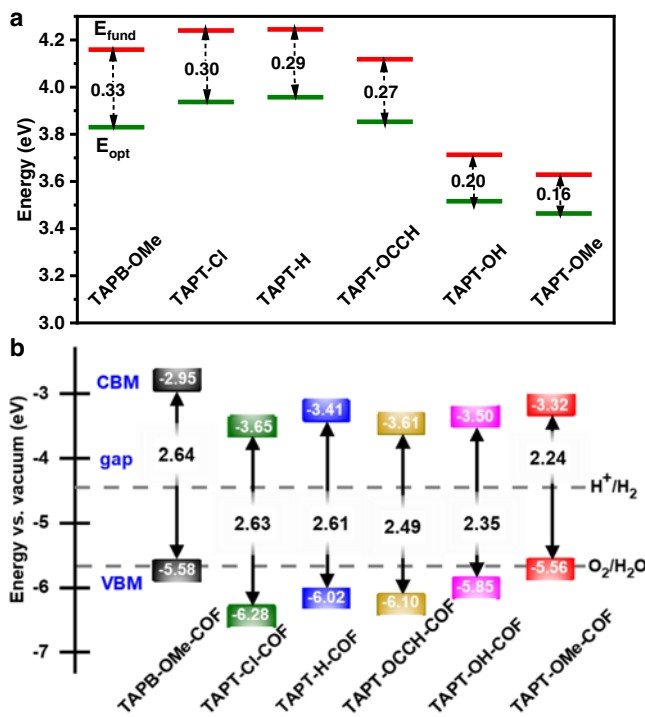

**Fig. 4 | Calculated exciton binding energy and band alignment of D-A pairs.** **a** Extrapolation results of fundamental gap ($E_{fund}$, red line) and optical gap ($E_{opt}$, green line) energies of D-A pairs. Inset: double arrows represent the exciton binding energy ($E_b = E_{fund} - E_{opt}$). **b** The calculated bandgap and band position of TAPB-OMe-COF (black), TAPT-Cl-COF (green), TAPT-H-COF (blue), TAPT-OCCH-COF (khaki), TAPT-OH-COF (purple) and TAPT-OMe-COF (red) relative to the vacuum level. The dashed lines are the redox potential of water at pH = 0.

of D-A interaction. Accordingly, even if TAPB-OMe and TAPT-OMe pairs differ just by their acceptors, the exciton binding energy of the former is around double of the latter (from 0.33 to 0.16 eV), because the former is the weakest yet the latter is the strongest regarding to the D-A interaction (Fig. 4a; Supplementary Fig. 4). Therefore, rational

selection of monomers based on the strength of D-A interaction is able to regulate the exciton binding energy.

## Band structures of D-A COFs
From the view of kinetic behavior of excitons, a small $E_b$ suggests the ease of exciton dissociation and effective separation of e-h pairs. Meanwhile, from the perspective of thermodynamics, the positions of conduction band minimum (CBM) and valence band maximum (VBM) in electronic band structures of COFs, should meet the requirement of half-reactions to make them take place (Fig. 4b; Supplementary Fig. 5)[9,46,47]. The CBM of all investigated COFs gives a more positive potential than the proton reduction potential (H⁺/H₂) of −4.44 eV (vs. vacuum, pH = 0), assuring the favorable energetics for electron transfer reaction from the COFs to proton. In fact, the TAPT-Cl-COF, TAPT-H-COF, TAPT-OCCH-COF and TAPT-OH-COF even exhibit the potential to achieve the water oxidation, due to the more negative VBM position than potential of water oxidation (−5.67 eV vs. vacuum, pH = 0). However, it is common to achieve only a half reaction experimentally with thermodynamically-allowed overall water splitting materials, due to the sluggish kinetics or other limitations[46].

## Synthesis and characterization of D-A COFs
With the aforementioned calculation results in mind, four COFs, namely TAPT-OMe-COF, TAPT-H-COF, TAPT-Cl-COF and TAPB-OMe-COF, as representatives have been experimentally fabricated[56–58]. The successful synthesis of these COFs is ensured by powder X-ray diffraction (XRD) patterns and solid-state cross-polarization magic angle spinning nuclear magnetic resonance (¹³C CP-MAS NMR) spectra (Fig. 5a; Supplementary Figs. 6 and 7). The formation of imine linkages has been demonstrated by the characteristic peaks at 1583, 1595, 1597 and 1616 cm⁻¹ for TAPT-OMe-COF, TAPT-H-COF, TAPT-Cl-COF and TAPB-OMe-COF, respectively, in the Fourier-transform infrared (FT-IR) spectra (Supplementary Fig. 8). N₂ adsorption for these COFs exhibits similar surface areas (1618–2708 m²/g) and pore sizes (2.5–2.7 nm), the latter of which are in consistent with the simulated results (Supplementary Figs. 9 and 10). All these COFs exhibit wide visible light absorption thanks to the π electron delocalization, and TAPT-OMe-COF shows an apparent red shift due to the strong D-A interaction

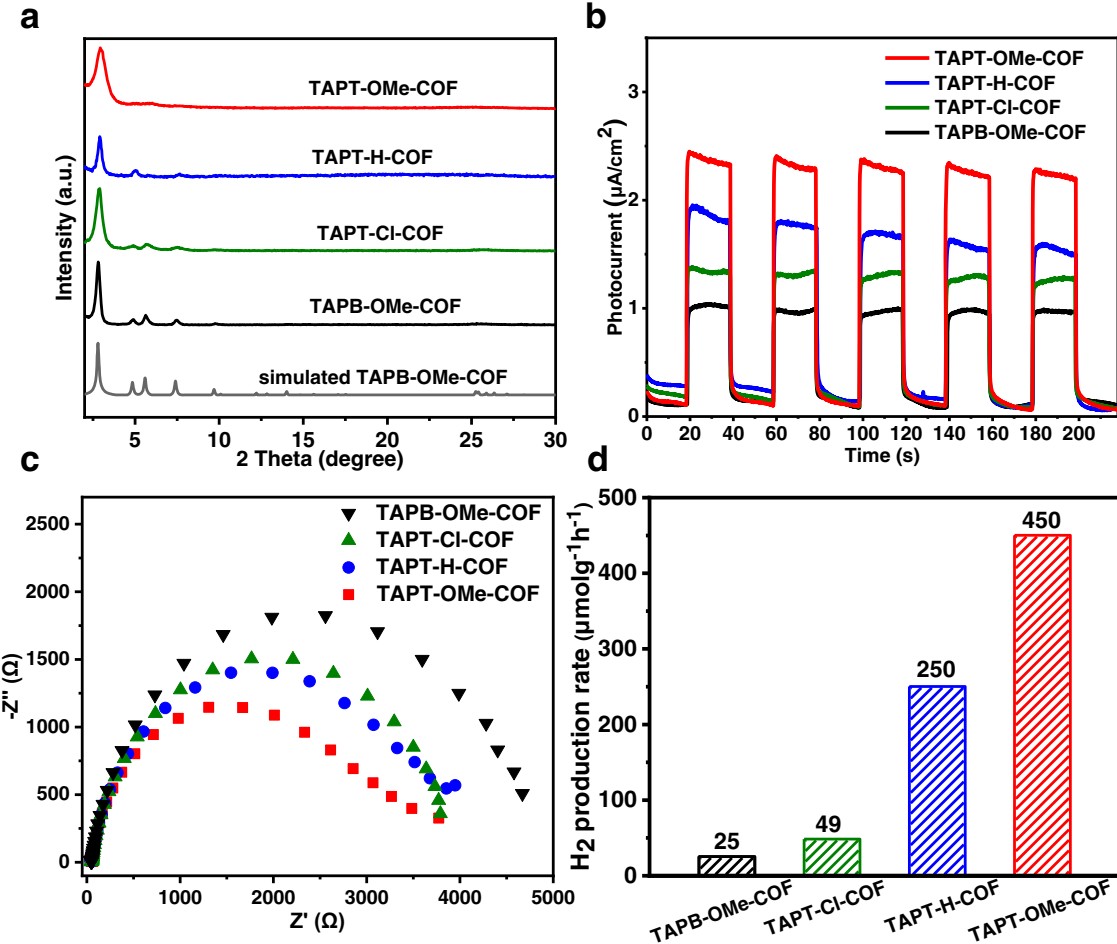

**Fig. 5 | Characterization and activity performance of D-A COFs. a** Powder XRD patterns. **b** Photocurrent responses. **c** EIS Nyquist plots. **d** Photocatalytic H$_2$ production rate. TAPB-OMe-COF, TAPT-Cl-COF, TAPT-H-COF and TAPT-OMe-COF are represented by black, green, blue and red colours, respectively.

which is consistent with the previous report (Supplementary Fig. 11)[59]. Mott-Schottky and Tauc plots are implemented to get the energy level of these COFs (Supplementary Figs. 12–15). The flat band positions (LUMO) of TAPT-OMe-COF, TAPT-H-COF, TAPT-Cl-COF and TAPB-OMe-COF are −0.66, −0.71, −0.85 and −0.77 V (*vs.* NHE, pH = 6.8), respectively, indicating their sufficient reducing power to achieve hydrogen production.

To reveal the amounts of photo-generated excitons and efficiency of exciton dissociation, photocurrent measurements have been conducted and the results show that TAPT-OMe-COF has the strongest photocurrent response, indicating the most effective charge transfer in all investigated COFs (Fig. 5b). This argument is supported by electrochemical impedance spectroscopy (EIS) results, where the radius of Nyquist plots follows a trend of TAPT-OMe-COF < TAPT-H-COF < TAPT-Cl-COF < TAPB-OMe-COF (Fig. 5c), and in consistent with the calculated order of $E_b$, revealing the efficient charge separation in TAPT-OMe due to the low charge transfer resistance. In addition, these results are also supported by photoluminescence (PL) emission spectroscopy, which offers important information for photo-induced charge transfer and recombination, further supporting the excitonic effects in these COFs[10]. The singlet excitons in TAPT-OMe-COF readily get dissociation resulting in the weakest PL emission whereas TAPB-OMe-COF features the strongest PL due to its very strong excitonic effects (Supplementary Fig. 16). The time resolved photoluminescence emission measurements suggest the mean PL lifetimes of 3.49, 1.69, 0.80 and 0.25 ns for TAPT-OMe-COF, TAPT-H-COF, TAPT-Cl-COF and TAPB-OMe-COF, respectively (Supplementary Fig. 17), which is

consistent with the increased trend of $E_b$[9,60]. The distinctly different optical and electrochemical properties among the COFs infer their difference in photocatalysis and the highest activity of TAPT-OMe-COF is assumed.

**Photocatalytic hydrogen production for D-A COFs**

We then set out to investigate the photocatalytic hydrogen production of the above COFs by water splitting under visible light irradiation, with TEA as a sacrificial agent and ~2 wt% Pt as co-catalyst (Supplementary Table 6). As expected, TAPT-OMe-COF presents the highest H$_2$ production rate of 450 μmol g$^{-1}$h$^{-1}$, far surpassing the other COFs with activity for TAPT-H-COF, TAPT-Cl-COF and TAPB-OMe-COF of 250, 49 and 25 μmol g$^{-1}$h$^{-1}$, respectively (Fig. 5d). Such an activity order is exactly the same as both the computed strength of D-A pairs and the trend of exciton binding energy. According to previous findings[33,35,36], the structure and energy levels of COFs are considered to be the main factors influencing photocatalysis. In this work, the four D-A COFs have similar structures and energy levels (Supplementary Figs. 11–15). In addition, other minor factors affecting the photocatalysis have also been investigated and demonstrated to be similar for the four COFs, such as hydrophilicity/dispersibility of the COFs (Supplementary Fig. 18) and sizes/loadings of the Pt co-catalyst (Supplementary Fig. 19; Supplementary Table 6). Based on these results, $E_b$ is assumed to be the dominant factor for the photocatalytic activity difference.

The stability of these COF photocatalysts has been examined by powder XRD after reaction (Supplementary Fig. 20). Unfortunately, the crystallinity of all these COFs significantly loses and most of their

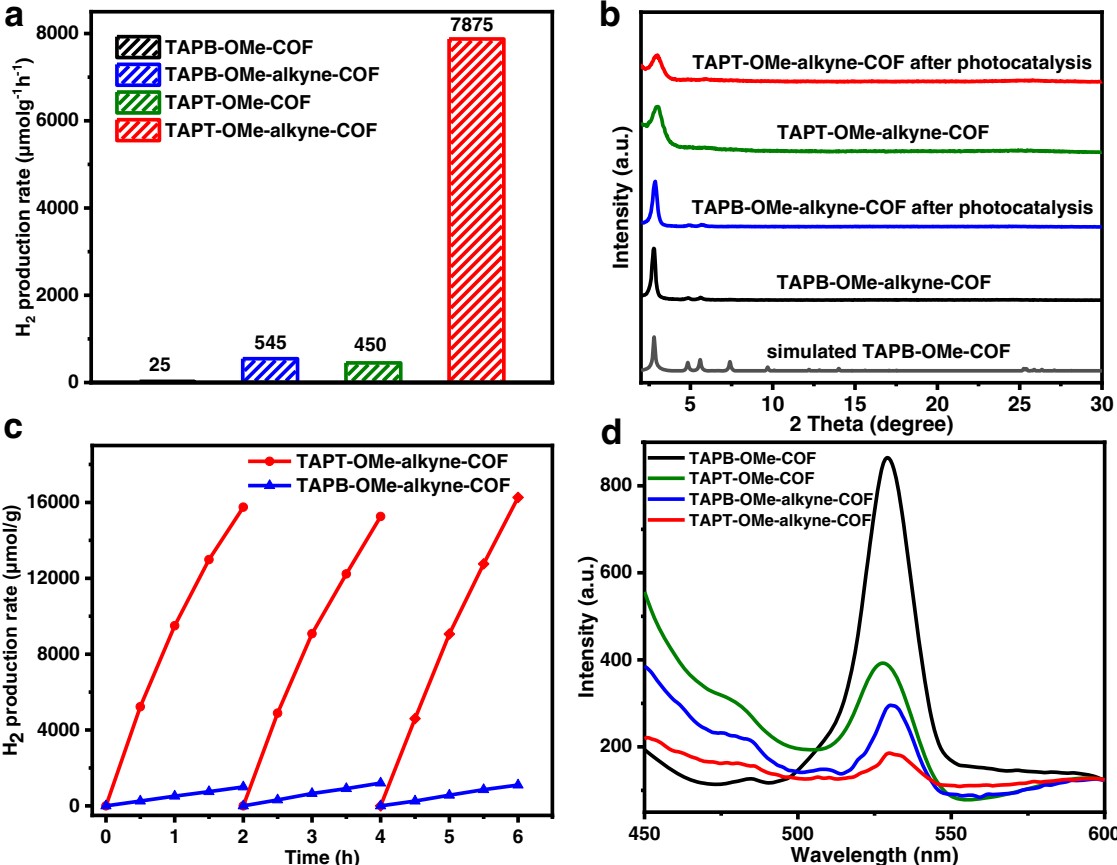

**Fig. 6 | Activity performance and fluorescence spectra of D-A COFs after modification. a** Photocatalytic $H_2$ production rate. **b** Powder XRD patterns after the photocatalytic $H_2$ production. **c** Recycling performance in photocatalytic $H_2$ production. **d** Steady-state fluorescence emission under excitation at 380 nm. The TAPB-OMe-COF, TAPT-OMe-COF, TAPB-OMe-alkyne-COF and TAPT-OMe-alkyne-COF are represented by black, green, blue and red colours, respectively.

frameworks completely collapse. This implies that the COF skeletons featuring reversible imine linkage are very likely to be broken under the photocatalytic hydrogen production conditions, resulting in the partial or complete structural damage.

### Synthesis and characterization of post-synthetic modification of D-A COFs

To improve the stability of imine COFs, post-synthetic modification of the worst- and best-performance COFs, TAPB-OMe-COF and TAPT-OMe-COF, has been selected to convert the imine linkage with phenylacetylene via the aza-Diels-Alder cycloaddition reaction[61,62], to yield TAPB-OMe-alkyne-COF and TAPT-OMe-alkyne-COF, respectively (Supplementary Fig. 21). The IR spectra provide significant evidence for this conversion, in which the $C = N$ peak intensity of TAPB-OMe-COF at 1616 $cm^{-1}$ disappears after the modification and two emerging new peaks at 1574 and 1544 $cm^{-1}$ for TAPB-OMe-alkyne-COF and 1571 and 1549 $cm^{-1}$ for TAPT-OMe-alkyne-COF are assignable to the characteristic peaks of quinolyl moiety (Supplementary Fig. 22)[62]. The apparent up-field shift of the triazine core and $-C = N$ resonances compared to the pristine COFs in $^{13}C$ CP-MAS NMR further support the successful post-modification (Supplementary Fig. 23)[61]. Powder XRD profiles and $N_2$ adsorption tests approve the retained crystalline structures for both alkyne-modified COFs (Supplementary Figs. 24–27). Both TAPB-OMe-alkyne-COF and TAPT-OMe-alkyne-COF exhibit widened visible light adsorption compared with the parent COFs, thanks to the extended π electron delocalization (Supplementary Fig. 28). The energy diagram of HOMO and LUMO levels for the alkyne-modified COFs can be delineated based on the Mott-Schottky and Tauc plots (Supplementary Figs. 29 and 30), where both

LUMO levels are more negative than the potential of hydrogen production, supporting their capability of photocatalytic hydrogen evolution.

### Photocatalytic hydrogen production for D-A COFs after modification

After the successful modification with phenylacetylene, the photocatalytic hydrogen production activity and stability for TAPB-OMe-alkyne-COF and TAPT-OMe-alkyne-COF have been investigated (Fig. 6a; Supplementary Table 7). Strikingly, TAPB-OMe-alkyne-COF and TAPT-OMe-alkyne-COF exhibit exponentially improved activities, respectively achieving 545 and 7875 $\mu mol\ g^{-1}h^{-1}$, which are ~18 times compared with the parent TAPB-OMe-COF and TAPT-OMe-COF. More importantly, in sharp contrast to the parent COFs, the stability of both modified COFs has been significantly improved under the same photocatalytic conditions thanks to the stable quinolyl linkage (Fig. 6b). Further recycling experiments for these two COFs suggest a negligible activity drop in the consecutive 3 runs of reaction (Fig. 6c). Transmission electron microscopy (TEM) observation demonstrates that the Pt nanoparticles in situ deposited by light irradiation are uniformly dispersed and their aggregation does not take place after the 3 photocatalytic cycles, due to the good restriction by the COF skeleton (Supplementary Figs. 31 and 32).

### Explanations of boosting activity for D-A COFs after modification

To decode the reason behind the much-enhanced activity after phenylacetylene modification, photoelectrochemical characterizations, including photocurrent, EIS and photoluminescence, have been also

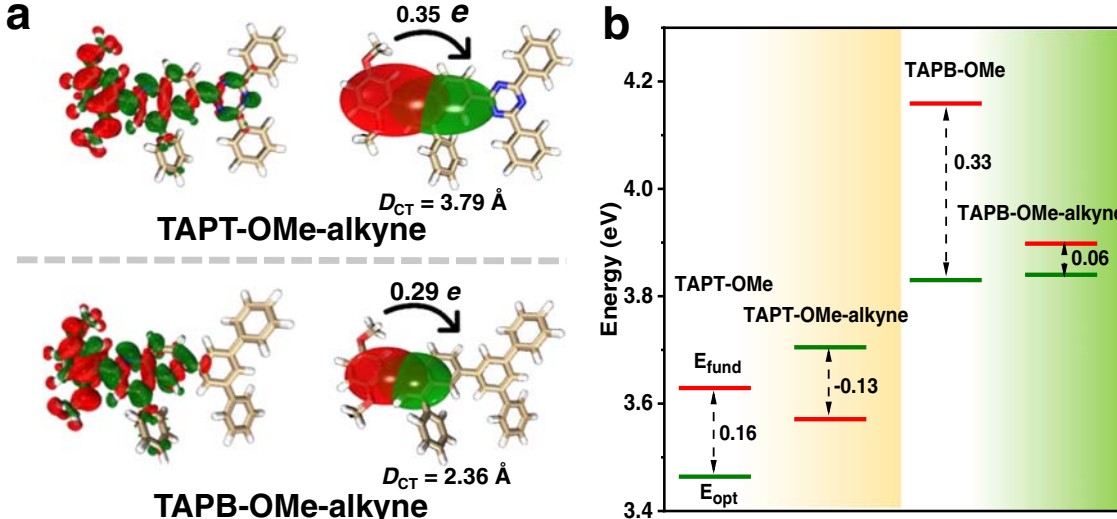

**Fig. 7 | DFT calculations to clarify the reasons for boosting activity after modification. a** Computed charge density difference between the ground and the excited states for TAPT-OMe-alkyne and TAPB-OMe-alkyne at an isovalue of 0.0005 a.u. (For each COF: $\Delta\rho$ is shown at the left side; centroids of charges are shown at the right side). The green and red represent increase and decrease in electron density, respectively. **b** Extrapolation results of fundamental gap ($E_{fund}$, red line) and optical gap ($E_{opt}$, green line) energies of TAPB-OMe and TAPT-OMe before and after the post-synthetic modification. Inset: double arrows represent the exciton binding energy ($E_b = E_{fund} - E_{opt}$).

carried out to unveil the charge separation efficiency. The steady-state PL emission for the COFs at ~530 nm arises from direct exciton recombination under 380 nm excitation at room temperature. The emission intensity for TAPB-OMe-alkyne-COF and TAPT-OMe-alkyne-COF is greatly weakened in comparison with the parent COFs, suggesting the strong suppression of radiative exciton recombination (Fig. 6d). In addition, both TAPB-OMe-alkyne-COF and TAPT-OMe-alkyne-COF provide the stronger photocurrent response and the lower charge transfer resistance than their respective parent COFs (Supplementary Figs. 33 and 34). These results corroborate each other that the recombination of photo-generated exciton is suppressed and exciton dissociation is promoted by post-synthetic modification with phenylacetylene, in which TAPT-OMe-alkyne-COF possesses the best charge transfer and separation, which are responsible for its highest activity in the photocatalytic $H_2$ production.

The degree of electron-hole separation evidently increases after the post-synthetic modification of TAPT-OMe and TAPB-OMe with phenylacetylene (Fig. 7a). Accordingly, the S/D value decreases (from 0.36 to 0.22) in TAPT-OMe and is even more remarkable (from 2.11 to 0.27) in TAPB-OMe after modification (Supplementary Table 8), indicating an improved charge transfer feature, possibly due to the formation of a larger conjugate system. Moreover, the larger $D_{CT}$ value and higher amount of transferred electron compared with the parent pairs suggest the stronger D-A interaction after modification. These give rise to an apparent decrease in the $E_b$ values of TAPT-OMe-alkyne and TAPB-OMe-alkyne, significantly down to −0.13 eV and 0.06 eV, respectively (Fig. 7b; Supplementary Fig. 35). The smaller the exciton binding energy is, it would be easier for the excitons to be dissociated into free carriers, exactly illustrating the boosting photocatalytic performance of the modified COFs.

Furthermore, temperature-dependent photoluminescence spectra have been conducted to estimate the $E_b$ experimentally for verifying the decreasing trend of calculated $E_b$ (Supplementary Figs. 36–39)[60]. The $E_b$ of TAPB-OMe-COF and TAPT-OMe-COF is evaluated to be 196 and 98 meV, respectively. After the post-modification, TAPB-OMe-alkyne-COF and TAPT-OMe-alkyne-COF give $E_b$ of 83 and 56 meV, much lower than the pristine COFs. The above results for modified COFs further verify the relationship that the reduced $E_b$ gives rise to improved photocatalytic activity.

## Further verification of the relationship between $E_b$ and photo-catalytic activity

To further validate the relationship between photocatalytic activity and $E_b$, the COF that does not fall in the D-A category has also been employed for investigation. The TAPB-H-COF, belonging to the D-D category, is selected as a representative. The DFT calculations indicate that the $E_b$ of TAPB-H (0.51 eV) is larger than all above D-A COFs, while TAPB-H-alkyne shows an evident decrease of $E_b$ to 0.26 eV, falling between TAPT-H (0.29 eV) and TAPT-OMe (0.16 eV). TAPB-H-COF and its alkyne-modified form, TAPB-H-alkyne-COF, have been successfully synthesized (Supplementary Fig. 40). The hydrogen production activity of 16 μmol g⁻¹h⁻¹ for TAPB-H-COF is lower than all above D-A COFs. Delightedly, TAPB-H-alkyne-COF exhibits a $H_2$ production rate of 307 μmol g⁻¹h⁻¹, falling between TAPT-H-COF and TAPT-OMe-COF, which is consistent with the order of $E_b$. These encouraging results suggest that the activity order could be predicted by the $E_b$, which might be not limited to the D-A COFs, as long as they have the similar structures and energy levels.

## Discussion

In summary, ten imine COFs made from two types of modules, i.e. amine and aldehyde monomers, have been evaluated by first-principles calculations. Seven of them bear the D-A framework in their minimal repeating structure motifs, which exhibit evidently different exciton binding energy, disclosing their potentially discriminated charge separation ability and photocatalytic performance. Directed by this computational guidance, four COFs (TAPB-OMe-COF, TAPT-Cl-COF, TAPT-H-COF and TAPT-OMe-COF) have been synthesized experimentally with the similar structures and energy levels. Their photocatalytic efficiency towards $H_2$ production by water splitting has been verified and the trend of their activities is in line with DFT predicted excitonic effects. To improve the stability of these COFs with reversible imine linkages, their post-synthetic modification with phenylacetylene affords the cycloaddition COFs. As expected, the resulting alkyne-modified COFs not only possess much improved stability but also manifest enhanced activity in exponential order and excellent recyclability toward photocatalytic $H_2$ production. Further calculation and experimental results by the temperature-dependent photoluminescence spectra suggest the very small exciton binding energy

for the COFs upon modification, well accounting for the promoted activity. This work not only for the first time achieves the rational fabrication of high-performance COF photocatalysts directed by the pre-design and exciton binding energy calculation with computational guidance, but also provides a research paradigm for advanced COF photocatalysis (from calculation prediction to experiments, then further stability and performance optimization with computational elucidation). Currently, while inorganic semiconductors are extensively studied due to their numerous advantages, COFs offer a unique advantage in precise fabrication and molecular-level structural regulation, which allow for mechanism elucidation and activity improvement. Therefore, COFs hold great potential for advanced photocatalysis in the future. Studies along this line by intelligent computation tools through machine learning for high-performance COF photocatalysts are underway in our laboratory.

## Methods

### Synthesis of TAPB-OMe-COF and TAPB-H-COF

The COFs were synthesized following the previous reports with some minor modifications[56]. Typically, TAPB (0.08 mmol, 28.1 mg) and 2,5-dimethoxyterephthalaldehyde (simply as Dma, 0.12 mmol, 23.3 mg) in a mixed solvent of o-dichlorobenzene/n-butanol/6 M HAc (0.5 mL/0.5 mL/0.1 mL) were frozen by liquid nitrogen in a glass tube. Then, the tube was evacuated, sealed off by the flame and stayed in the oven at 120 °C for 3 days. The yellow product was collected by centrifugation and thoroughly washed with THF and methanol. Then the samples were activated by supercritical $CO_2$ prior to the further use. The synthesis of TAPB-H-COF is the same as synthesis of TAPB-OMe-COF except the substitution of Dma (23.3 mg) to 1,4-phthalaldehyde (15.9 mg).

### Synthesis of TAPT-Cl-COF

Specifically, TAPT (0.08 mmol, 28.4 mg) and 2,5-dichloroterephthalaldehyde (0.12 mmol, 24.4 mg) in a mixed solvent of o-mesitylene/1,4-dioxane/6 M HAc (0.8 mL/0.2 mL/0.1 mL) were frozen by liquid nitrogen in a glass tube. Then, the tube was evacuated, sealed off by the flame and stayed in the oven at 120 °C for 3 days. The subsequent processing steps are the same as those for the above preparation of TAPB-OMe-COF.

### Synthesis of TAPT-H-COF and TAPT-OMe-COF

Typically, TAPT (0.06 mmol, 21.3 mg) and 1,4-phthalaldehyde (0.09 mmol, 12.1 mg) or Dma (0.09 mmol, 17.5 mg) in a mixed solvent of o-dichlorobenzene/n-butanol/6 M HAc(1 mL/1 mL/0.2 mL) were frozen by liquid nitrogen in a glass tube. Then, the tube was evacuated, sealed off by the flame and stayed in the oven at 120 °C for 3 days. The subsequent processing steps are the same as those for the above preparation of TAPB-OMe-COF.

### Post-synthetic modification of the D-A COFs

The synthesis was according to the literatures with a bit modification[61]. Specifically, 20 mg TAPB-OMe-COF or TAPT-OMe-COF, 30 μL phenylacetylene, 40 mg chloranil, 20 μL BF$_3$•OEt$_2$ and 10 mL toluene were added into a 25 mL Schlenk tube. The tube was degassed and sealed off by three freeze-pump-thaw and heated at 110 °C in oil bath under Argon atmosphere for 3 days. After cooling down, the mixture was isolated via centrifugation. The precipitate was washed with DMF for 3 times, and quenched with saturated NaHCO$_3$ for two times. The COFs were continuously washed with water for more than 10 times to remove the residual NaHCO$_3$ thoroughly. Subsequently, the solids were washed with DMF and CH$_3$OH. Finally, the samples were activated by supercritical $CO_2$ prior to the further use.

### Photocatalytic hydrogen production

The photocatalytic hydrogen production experiments were performed with 5 mg catalysts in 10 mL CH$_3$CN, 0.2 mL deionized water and 4 μL H$_2$PtCl$_6$ aqueous solution (37.5 mg/mL, actual amount: 2 wt% Pt determined by ICP-AES result). The mixture was dispersed by ultrasonication and put into a 160 mL optical reaction vessel (Beijing Perfectlight Technology Co., LTD, China). The vessel was purged with nitrogen for 10 min to remove the air. After that, 0.5 mL triethylamine (TEA) was injected to the reactor. The light source is a 300 W Xe lamp equipped with a UV cut-off filter (>380 nm). The light intensity at the bottom of the reactor was -3.0 W/cm$^2$ obtained by VLP-2000 optical laser power meter. During the reaction, the circulating water is to ensure the constant reaction temperature at about 25 °C. The rotation speed of the stir bar is kept at 300 rpm. For each evaluation of hydrogen generation, 400 μL of the headspace was injected into the gas chromatography (Shimadzu GC-2014, argon as a carrier gas) using a thermal conductivity detector (TCD) and quantified by a calibration plot to the internal hydrogen standard.

### Computational details

All calculations of molecular models were performed with Gaussian 16 package[63]. M06-2X functional and the 6−31 G (d,p) basis set was employed for the calculations of IP, EA, $E_{opt}$, and $E_b$ energies (in molecular geometries optimized with ωB97XD/6-31 G(d,p))[16,64]. The effect of solvation by acetonitrile was considered by using the Polarizable Continuum Model (PCM) solvation model[65].

The simulation of periodic COF structure was carried out using the Vienna ab initio Simulation Package (VASP) code[66], with the Perdew-Burke-Ernzerhof exchange-correlation functional[67]. The DFT-D3(BJ) dispersion correction was utilized for the consideration of the van der Waals (vdW) interactions[68]. The Brillouin zones were sampled with $3 \times 3 \times 1$ Γ-centered k-points. The plane-wave basis set within a kinetic-energy cutoff of 400 eV was used for describing the wave function. The hybrid functional of HSE06 which combines HF exchange with the exchanges of PBE exchange-correlation functional, was employed for the calculation of electronic structure[69]. A vacuum region of about 15 Å was used to eliminate the interaction of layers for 2D imine-linked COFs. The convergence criterion of forces during atomic optimizations was set to 0.02 eV Å$^{-1}$. The energy convergence criterion in the self-consistent field iteration was $10^{-4}$ eV in optimizations while $10^{-5}$ eV in static calculations. More details of the DFT calculation are listed in the supplementary information.

## Data availability

All data are available in the main text or the supplementary information. Source data are provided with this paper.

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

## Acknowledgements

Y.Q. and Y.H. contributed equally to this work. H.-L.J. is thankful for the National Key Research and Development Program of China (2021YFA1500400), the Strategic Priority Research Program of the CAS (XDB0450302), the NSFC (22161142001, U22A20401), and International Partnership Program of CAS (123GJHZ2022028MI). G.Z. thanks the start-up funding of University of Science and Technology of China (KY2340000151).

## Author contributions

H.-L.J. conceived and designed the project. Y.Q. performed the experiments and collected the data. Y.H. and G.Z. conducted the DFT calculations and analyzed the data. X.Z. and G.Y. participated in some experiments. H.-L.J., Y.Q. and Y.H. co-wrote the manuscript. All authors discussed the results and commented on the manuscript.

## Competing interests

The authors declare no competing interests.
