## [Peer Review File · Nature Communications]

Computation-Based Regulation of Excitonic Effects in Donor-Acceptor Covalent Organic Frameworks for Enhanced PhotocatalysisREVIEWER COMMENTS

Reviewer #1 (Remarks to the Author):

Qian, Han, Zhang, Yang, Zhang and Jiang tried to clarify the relationship between structure and exciton binding energy in donor-acceptor based covalent organic framework (COF).

In the present paper, the authors focused on the COF based on TAPB/TAPT as acceptor group and terephthaldehyde derivatives as donor group. The energy levels of the groups have been adjusted by the aspects derived from theoretical calculations, besides, the nature of charge transfer by several computational approach. The trends were also captured experimentally, suggesting the theoretical predictions as successful for fine tuning of photon (light) energy harvesting. Modification of the two COFs further improved their stability and H₂ production rate: the clear feedbacks from the computation. "Computation-Guided Regulation": apparently the highlight of the present work as the authors are claiming in the title. The reviewer agrees the computational protocols are successful for the design of the COF system, however feels simultaneously the claims are too much after reading this manuscript. Most of all the readers may be disappointed by the story in contrast to the overstated title. The examined pairs of donors and acceptors are, as we see in the papers in the list of references, can be demonstrated even in one isolated paper, and one can realise (has realised) that the prime factor impacting on the overall outcomes as photocatalysts is rather than the excitonic effects in the actual systems. Thus I feel the impact of this paper in the field is a bit short and limited, and I am hesitating to recommend the paper in Nature Communications with extremely wide readership. Followings are some critical issues the authors should address in case of submission to the other journals or resubmission to this journal. In particular, the point two is substantial; I cannot recommend publication of this article unless addressing this point.

Major comments:

1. In page 3 line 5, I don't recommend to use the term "oriented movement" in this context because both direct formation of excitation by electronic transition and indirect formation by charge transfer process are dominated by probability and its direction is also not determined, not oriented.
2. How did the authors generate the isosurface in Fig 3? For me, they look all ellipsoid, it won't be in such a shape. If it is wrong, they are quite misleading. The authors already calculated in DFT-level, so the difference in density can be accurately generated. (same for Fig7)
3. Please provide simulated XRD patterns for other samples including AA stacking and AB stacking in at least supporting information.
4. I could not find the details of device structure and its fabrication for the photocurrent measurements. Will the absorption coefficient or the thickness affect the photocurrent results? (same for impedance spectroscopy.)
5. Please add the reaction scheme for post-modification in SI.
6. Similar D-A type COF strategies have been reported recently. The authors may consider to cite them, for example, <https://pubs.acs.org/doi/10.1021/acscatal.2c02173>

Minor comments:

1. Page 12, line 16, chozen would be chosen.

Reviewer #2 (Remarks to the Author):

This paper reports the impact of exciton binding energy of donor-acceptor covalent organic frameworks in photocatalytic H₂ production. The authors provide a detailed description on how the exciton binding energy can be regulated to increase the photocatalytic efficiency. The theoretical results are compared with experiments demonstrating an excellent correlation. In addition, the issue of process stability was addressed, where an interesting and simple molecular modification in the acceptor component was successfully employed. Interestingly, the molecular modification also led to a significant increase in the H₂ production rate. The possibility of using the theory to guide the synthesis process of the materials completely developed in this work proved to be very efficient. The manuscript is well written, easy to follow, and it does improve our understanding about the correlation between exciton binding energy and photocatalytic H₂ production, in my judgment. I enjoyed reading the paper and recommend it for publication. I have a few comments (see below) that the authors should take into account.

Minor remark:

1. In my opinion, the first paragraph of the manuscript lacks mention of the state of the art production of H₂. Likewise, at the end of the manuscript it is necessary to correlate the results obtained with the state of the art.
2. In the same context as my first comment, is there any advantage of the method used in this report to produce H₂ over other methods? For example, with regards to complexity.
3. In the introduction, it is necessary to clearly mention theoretical works (DFT) in the area of organic solar cells that also use this strategy of weakening the exciton binding energy of molecules to optimize the photovoltaic process (<https://doi.org/10.1021/acs.jpcc.8b12261>, <https://doi.org/10.1021/acs.jpcc.8b07197>, <https://doi.org/10.1039/C9TC03563J>). Furthermore, in the introduction it is mentioned that "...D-A molecular junctions inspired by solar cells have received intensive attention due to easy construction and structural diversity." If I'm not mistaken, the three references cited in this sentence do not correspond to the statement.
4. Some figures (1-4) are not in good resolution and should be improved.
- 5- It would be interesting to specify the functional and basis of DFT calculations also in the

supplementary information.

6. There are two references 46, this should be fixed with renumbering.

7. Does the dihedral angle between the D and A components of the studied molecules have any correlation with the exciton binding energy?

Response to Reviewer 1:

General Comment: “Qian, Han, Zhang, Yang, Zhang and Jiang tried to clarify the relationship between structure and exciton binding energy in donor-acceptor based covalent organic framework (COF).

In the present paper, the authors focused on the COF based on TAPB/TAPT as acceptor group and terephthaldehyde derivatives as donor group. The energy levels of the groups have been adjusted by the aspects derived from theoretical calculations, besides, the nature of charge transfer by several computational approach. The trends were also captured experimentally, suggesting the theoretical predictions as successful for fine tuning of photon (light) energy harvesting. Modification of the two COFs further improved their stability and H₂ production rate: the clear feedbacks from the computation. “Computation-Guided Regulation”: apparently the highlight of the present work as the authors are claiming in the title. The reviewer agrees the computational protocols are successful for the design of the COF system, however feels simultaneously the claims are too much after reading this manuscript. Most of all the readers may be disappointed by the story in contrast to the overstated title. The examined pairs of donors and acceptors are, as we see in the papers in the list of references, can be demonstrated even in one isolated paper, and one can realise (has realised) that the prime factor impacting on the overall outcomes as photocatalysts is rather than the excitonic effects in the actual systems. Thus I feel the impact of this paper in the field is a bit short and limited, and I am hesitating to recommend the paper in Nature Communications with extremely wide readership. Followings are some critical issues the authors should address in case of submission to the other journals or resubmission to this journal. In particular, the point two is substantial; I cannot recommend publication of this article unless addressing this point.”

Response: We much appreciate the comments from the reviewer. In response to the questions raised, we have made revisions and provided corresponding explanations below.

1) Thanks for your kind concern about the title. After re-assessing the content of the work and its title, we have now revised it to “Computation-Based Regulation of Excitonic Effects in Donor-Acceptor Covalent Organic Frameworks for Enhanced Photocatalysis”, toning down the claim and making it easier for readers to understand the main purpose of the work.

2) Yes, there are indeed many factors influencing the performance of photocatalysis in D-A COFs. In this work, we aim to control the excitonic effect as the dominant factor, and fix all other possible variables in the photocatalysts investigated (the four D-A COFs). Previous reports (see refs: *Appl. Catal. B* **2020**, 276, 119174; *Chem. Soc. Rev.* **2020**, 49, 4135) have indicated that, the structure and energy levels of COFs are the primary factors affecting photocatalysis. It is apparent that the four representative D-A COFs share similar structures and energy levels (with a bandgap between 2.46 and 2.64 eV, and a LUMO level between -0.66 and -0.85 eV) (see SI: Figures S11-S15). **In this case (for the photocatalysts with similar structures and energy levels), how would excitonic effects (or E_b) regulate photocatalysis? This is just what we would present to readers in this work.**

In fact, in addition to the similar structures and energy levels of the four COFs, the following other minor factors possibly influencing photocatalytic performance have been considered: a) The static water contact angles of the four COFs, are in the range of 83°-96° (see SI: Figure S18), indicating their similar hydrophilicity and dispersibility. b) The sizes and loadings of Pt co-catalyst in the four COF photocatalysts are also similar (see SI: Figure S19, Table S6).

Related discussion on the excitonic effect as the prime/dominant factor impacting on the overall photocatalytic performance has been added in text and SI (see text: page 13, the 1st paragraph, line 4-

11; see SI: Figure S18-19).

3) The important features concerning the results reported in this work have been summarized as follows, which, we believe, are of great interest to general readers of *Nature Commun.*:

a) In comparison with polymeric conductors, COFs facilitate precise fabrication and structural tailoring, giving the opportunity to accurately control over excitons and disclose the relationship between the strength of D-A interactions and excitonic effects. **This work employs ideal photocatalysts (e.g., COFs) to investigate how D-A interaction strengths in COFs affect excitonic effects and the corresponding photocatalytic activity.**

b) Though the formation of D-A structure is frequently mentioned in COF photocatalysis, whether the donor and acceptor monomers can truly give a D-A structure remains unclear in COFs. In this work, **DFT calculations have been adopted to estimate the feasibility of D-A COF formation.** Moreover, for a series of potential D-A COFs, **the D-A interaction strength and exciton binding energy (E_b), as well as the corresponding photocatalytic activity trend, have been theoretically predicted.**

c) According to the calculation results, seven D-A COFs can be formed out of ten possible combinations, in which the predicted D-A COF, *i.e.* TAPT-OMe-COF, featuring the strongest D-A strength yet the lowest E_b , would present the highest photocatalytic activity. **This computational prediction makes it dispensable for a large number of experimental trials to the synthesis of COFs with ordinary performance, directly screening out the best-performance COF.**

d) Four D-A COFs have been synthesized and their activity trend in the photocatalytic H_2 production is in good agreement with the prediction by the calculation. Furthermore, the post-modification of the imine linkage to quinoline moiety in these D-A COFs enables them higher stability and much enhanced photocatalytic activity up to $7875 \mu\text{mol h}^{-1}\text{g}^{-1}$ due to the significantly reduced E_b . **This work provides an effective strategy for improving photocatalytic activity, stability, and recyclability for imine COFs.**

e) This work not only for the first time achieves the rational fabrication of high-performance COF photocatalysts directed by the pre-design and exciton binding energy calculation with computation guidance, but also **provide a new research paradigm for advanced COF photocatalysis (from calculation to experiments, then further stability and performance optimization with computational elucidation).**

Given the significance of our findings, we believe that this work will have a profound impact in the field of COF/polymer photocatalysis, and will encourage readers to pay greater attention to the impact of excitonic effects on photocatalysis, an area that has been long ignored.

Major comments:

Comment 1: *"In page 3 line 5, I don't recommend to use the term "oriented movement" in this context because both direct formation of exciton by electronic transition and indirect formation by charge transfer process are dominated by probability and its direction is also not determined, not oriented."*

Response: Many thanks for the very kind reminder. As per your suggestion, we have removed the term 'oriented movement'. Related revisions have been made in the text (see text: page 3, the 1st paragraph, line 2 from the bottom).

Comment 2: *"How did the authors generate the isosurface in Fig 3? For me, they look all ellipsoid, it won't be in such a shape. If it is wrong, they are quite misleading. The authors already calculated in DFT-level, so the difference in density can be accurately generated. (same for Fig7)?"*

Response: Thanks a lot for the professional comments. The form of $\Delta\rho$ (see text: Figure 3, the left-side image for each COF), as requested by the reviewer, presents the original charge density difference between the ground states (GS) and the excited states (ES). In fact, the form of centroids of charges (see text: Figure 3, the right-side image for each COF), which we assume to give a more visual and intuitive presentation of the density difference between the GS and ES, is **just another form of representation for $\Delta\rho$** . The centroids of charge employs a method described for the computed charge difference in a previous theoretical study (*J. Chem. Theory Comput.* **2011**, *7*, 2498), which has been well accepted and cited over 800 times. This method has also been employed in many previous studies (e.g., *Angew. Chem. Int. Ed.* **2023**, e202300256; *Nat. Commun.* **2021**, *12*, 320; *Acc. Chem. Res.* **2022**, *55*, 2698; *Chem. Sci.* **2020**, *11*, 3405) and it can be considered as a useful method to simplify charge density difference ($\Delta\rho$).

Specifically, the form of $\Delta\rho$ was generated using the following formula:

$$\Delta\rho(r) = \rho_{\text{ES}}(r) - \rho_{\text{GS}}(r) \quad (1)$$

The form of centroids of charges defines two centroids of charges (C_+ and C_-) associated with the positive and negative density regions. Firstly, the root-mean-square deviations along the three axes ($\sigma_{aj}, j = x, y, z; a = + \text{ or } -$) are computed as

$$\sigma_{aj} = \sqrt{\frac{\sum_i \rho_a(r_i)(j_i - j_a)^2}{\sum_i \rho_a(r_i)}} \quad (2)$$

Then the two centroids can then simply be defined as

$$C_+(r) = A_+ e \left(-\frac{(x-x_+)^2}{2\sigma_{+x}^2} - \frac{(y-y_+)^2}{2\sigma_{+y}^2} - \frac{(z-z_+)^2}{2\sigma_{+z}^2} \right) \quad (3)$$

$$C_-(r) = A_- e \left(-\frac{(x-x_-)^2}{2\sigma_{-x}^2} - \frac{(y-y_-)^2}{2\sigma_{-y}^2} - \frac{(z-z_-)^2}{2\sigma_{-z}^2} \right) \quad (4)$$

For more detailed information, please refer to the aforementioned JCTC (*J. Chem. Theory Comput.* **2011**, *7*, 2498). The structures of these centroids are obtained from Gaussian functions, where the values asymptotically approach zero from the centroid of the function, resembling the shape of an ellipsoid as concerned by the reviewer. The above-mentioned calculation method has been supplemented in the SI (see SI: page 7-8, S4 Supplementary Calculation Details).

Clearly, the form of $\Delta\rho$ appears to be less intuitive than centroids of charges, due to the intertwining of positive and negative charge densities, hindering the identification of actual donor and acceptor moieties as well as the comparison of the extent of charge transfer between different substituents. By contrast, the computed centroids, which involve the use of C_+ and C_- functions, offer a clearer image, enabling better discernment of the spatial characteristics of charge transfer between the donor and acceptor. In addition, the distance of charge transfer (D_{CT}), which is calculated from the total charge transfer length between the negative and positive barycenters, is better represented through this form. In response to your concerns, we have included the suggested form of charge density difference ($\Delta\rho$) in the Figure 3 and 7 (see text: Figure 3 and 7a). Related revisions have been made in the text and SI (see text: page 8, the 1st paragraph from the bottom, line 1-4 and Figure 3; page 17, Figure 7).

Comment 3: "Please provide simulated XRD patterns for other samples including AA stacking and AB stacking in at least supporting information."

Response: We are thankful for the constructive suggestion. The simulated XRD patterns including AA stacking and AB stacking for D-A COFs (see SI: Figure S6) and post- synthetic modified COFs (see SI:

Figure S25) have been added in the SI.

Comment 4: *"I could not find the details of device structure and its fabrication for the photocurrent measurements. Will the absorption coefficient or the thickness affect the photocurrent results? (same for impedance spectroscopy.)"*

Response: Many thanks for the kind concern. The experimental details of photocurrent and EIS measurement can be found in the SI (see SI: page 6, S3 Photoelectrochemical Measurements). In addition, the device structure and its fabrication for the photocurrent and EIS measurements have been also supplemented in the SI (see SI: Figure S41 and S42).

The photocurrent is primarily determined by the absorption of incident light (amounts of photo-generated excitons) and the efficiency of charge separation when fixing the light irradiation area. Consequently, the material's absorption coefficient and thickness, which are related to light absorption, also affect the photocurrent results. In this work, catalysts of identical weight were thoroughly dispersed in ethanol using ultrasonic apparatus, and the resulting solution was dropped to the FTO substrate, covering the same area. Therefore, the comparison of photocurrent results between catalysts should not be significantly influenced by variations in thickness.

The absorption coefficient of the photocatalysts can be estimated through the absorbance in UV-vis spectra (see SI: Figure S11). All the D-A COFs present similar absorption coefficients, with the exception of TAPT-H-COF, which is notably stronger than the others. However, TAPT-H-COF does not exhibit the strongest photocurrent response among the four COFs. The joint characterization (such as EIS), which is irrelevant with absorption coefficient, shows a similar trend to the photocurrent results, implying that charge separation efficiency is the determining factor, even if absorption coefficient gives influence on the photocurrent results to some extent. We have made revisions to the related statements for rigorous expression and in response to the reviewer's concerns (see text: page 12, the 2nd paragraph, line 1).

In impedance spectroscopy, the absorption coefficient does not appear to affect the results of EIS, while the thickness of the catalyst might have an impact. However, to ensure a fair comparison of EIS results, we coated the same amounts of catalysts on the same area of the electrode, resulting in similar thicknesses of D-A COFs that have minimal effect on the results.

Comment 5: *"Please add the reaction scheme for post-modification in SI."*

Response: Thanks a lot for the constructive suggestion. The reaction scheme for post-synthetic modification of COFs has been added in SI (see SI: Figure S21).

Comment 6: *"Similar D-A type COF strategies have been reported recently. The authors may consider to cite them, for example, <https://pubs.acs.org/doi/10.1021/acscatal.2c02173>"*

Response: Many thanks for the kind suggestion. We have added some important references regarding similar D-A type COF strategies into the text (see text: page 27, references 40-41).

Minor comments:

Comment 1: *"1. Page 12, line 16, chozen would be chosen."*

Response: Thanks for the valuable reminder. In response to the reviewer's general comment regarding variable control, we have reorganized this section (see text: page 13, the 1st paragraph, line 4-11). The chozen has been delated.

Response to Reviewer 2:

General Comment: *“This paper reports the impact of exciton binding energy of donor-acceptor covalent organic frameworks in photocatalytic H₂ production. The authors provide a detailed description on how the exciton binding energy can be regulated to increase the photocatalytic efficiency. The theoretical results are compared with experiments demonstrating an excellent correlation. In addition, the issue of process stability was addressed, where an interesting and simple molecular modification in the acceptor component was successfully employed. Interestingly, the molecular modification also led to a significant increase in the H₂ production rate. The possibility of using the theory to guide the synthesis process of the materials completely developed in this work proved to be very efficient. The manuscript is well written, easy to follow, and it does improve our understanding about the correlation between exciton binding energy and photocatalytic H₂ production, in my judgment. I enjoyed reading the paper and recommend it for publication. I have a few comments (see below) that the authors should take into account.”*

Response: We much appreciate the reviewer’s very positive evaluation and constructive suggestions.

Comment 1: *“In my opinion, the first paragraph of the manuscript lacks mention of the state of the art production of H₂. Likewise, at the end of the manuscript it is necessary to correlate the results obtained with the state of the art.”*

Response: We are thankful for the constructive comments. In the first paragraph, we have introduced the state-of-the-art production of H₂ from the perspective of photocatalysts. Additionally, at the end of the manuscript (in the Discussion section), the comparison between the results obtained and those of the current state-of-the-art photocatalysts have been introduced. Related revisions have been made in the text (see text: page 2, the last paragraph, line 2-5; page 19, line 3-6 from the bottom; page 23, references 3-4).

Comment 2: *“In the same context as my first comment, is there any advantage of the method used in this report to produce H₂ over other methods? For example, with regards to complexity.”*

Response: Thanks a lot for the valuable comments. In this work, compared to previous methods, the method employed in this work offers the following advantages:

- 1) From a research method perspective, this work employs DFT calculations to screen target photocatalysts using exciton binding energy as the primary index. Compared to traditional methods for photocatalysis research, this approach offers several advantages, including higher efficiency, reduced costs, and improved accuracy in predicting the performance of photocatalysts.
- 2) From a photocatalyst perspective, polymer semiconductors offer several advantages over traditional inorganic semiconductors, including not requiring metals, being low-toxic, inexpensive, and having adjustable band gaps. Especially for COFs, crystalline polymers with precise and tailorable structures, offer an advantageous platform for investigating the relationship between structure and photocatalytic activity through molecular-level structural regulation and hold great potential for advanced photocatalysis in hydrogen production.

Comment 3: *“In the introduction, it is necessary to clearly mention theoretical works (DFT) in the area of organic solar cells that also use this strategy of weakening the exciton binding energy of molecules to optimize the photovoltaic process (<https://doi.org/10.1021/acs.jpcc.8b12261>,*

<https://doi.org/10.1021/acs.jpcc.8b07197>, <https://doi.org/10.1039/C9TC03563J>). Furthermore, in the introduction it is mentioned that "...D-A molecular junctions inspired by solar cells have received intensive attention due to easy construction and structural diversity." If I'm not mistaken, the three references cited in this sentence do not correspond to the statement."

Response: Many thanks for the positive suggestions and valuable reminder. The use of D-A junctions from the theoretical works in the field of organic solar cells has been introduced in the text, with relevant references included (see text: references 16-18). Furthermore, the inconsistencies in the references have been rectified. Related revisions have been made in the text (see text: page 3, the 1st paragraph, line 4-7; page 24, references 16-18).

Comment 4: "Some figures (1-4) are not in good resolution and should be improved."

Response: Thanks a lot for the kind reminder. The resolution of Figures 1-4 has been adjusted for optimal clarity (see text: Figures 1-4).

Comment 5: "It would be interesting to specify the functional and basis of DFT calculations also in the supplementary information."

Response: Many thanks for the kind suggestion. The functional and basis of DFT calculations have been added in the supplementary calculation details (see SI: page 7, S4 Supplementary Calculation Details, the 1st paragraph).

Comment 6: "There are two references 46, this should be fixed with renumbering."

Response: Thanks for the valuable reminder. The errors identified in the reference have been corrected. Related revisions have been made in the text (see text: page 29, reference 64).

Comment 7: "Does the dihedral angle between the D and A components of the studied molecules have any correlation with the exciton binding energy?"

Response: We are thankful for the interesting and nice direction. In our work, the studied molecules (D-A pairs) containing TAPT_c part (TAPT-Cl, TAPT-H, TAPT-OCCH, TAPT-OH and TAPT-OMe) are planar (a dihedral angle of 0 degrees), while only TAPB-OMe has a dihedral angle of 43.3 degrees (see text: Figure 3). Given that TAPB-OMe exhibits the largest dihedral angle and exciton binding energy, it is reasonable to assume that a smaller dihedral angle would correspond to a decreased exciton binding energy. However, the limited number of examples of dihedral angles in this study makes it difficult to draw a reliable conclusion.

Future work will further examine the relationship between the dihedral angle and exciton binding energy in COFs, but this would be a substantial undertaking and merit a manuscript in its own right.

REVIEWERS' COMMENTS

Reviewer #1 (Remarks to the Author):

The revised title of the manuscript becomes much better than the initial one, away from misleading of the readers. Still the term of “computation” bothers me because one can lead the choice of the chromophores among the building blocks of COFs by “thinking” without computations. It might be acceptable in the field of general chemistry.

The major changes made in this revised version of manuscript are in my comments 2 for the initial submission; the protocol to lead isosurface of figure 3. Indeed the use of differential charge density as discussed in J. Chem. Theory Comput. 2011 have given a better interpretation phenomenologically as the authors are referring the papers in ACIE, Nat. Commun. ACR, and Chem Sci. The last one is discussing on a bit shifted system from the ones in the present study, however minding us some new insights of experiments: electron paramagnetic resonance under photo-illumination. The system will provide a clear answer for the authors hypothesis of “computation guided design” of the system quantitatively by the clear correlation between the equilibrium yield of the primary intermediates and the choice of chromophores. It is, however, another story.

Overall I feel the manuscript is now ready for publication in Nature Communications.

Reviewer #2 (Remarks to the Author):

The authors have implemented all suggestions and I believe the manuscript is in good shape for publication. For this reason, I congratulate the authors for the excellent work done.

Dr. Hai-Long Jiang, Professor, FRSC
Department of Chemistry
Hefei National Research Center for Physical Sciences at the Microscale
University of Science and Technology of China (USTC)

Response to Reviewer 1:

General Comment: *"The revised title of the manuscript becomes much better than the initial one, away from misleading of the readers. Still the term of "computation" bothers me cause one can lead the choice of the chromophores among the building blocks of COFs by "thinking" without computations. It might be acceptable in the field of general chemistry.*

*The major changes made in this revised version of manuscript are in my comments 2 for the initial submission; the protocol to lead isosurface of figure 3. Indeed the use of differential charge density as discussed in *J. Chem. Theory Comput.* 2011 have given a better interpretation phenomenologically as the authors are referring the papers in *ACIE*, *Nat. Commun.*, *ACR*, and *Chem Sci*. The last one is discussing on a bit shifted system from the ones in the present study, however minding us some new insights of experiments: electron paramagnetic resonance under photo-illumination. The system will provide a clear answer for the authors hypothesis of "computation guided design" of the system quantitatively by the clear correlation between the equilibrium yield of the primary intermediates and the choice of chromophores. It is, however, another story.*

*Overall I feel the manuscript is now ready for publication in *Nature Communications*."*

Response: We very much appreciate the positive comments. Specifically,

- 1) Thanks a lot for your kind approval of the revised title. As per your suggestion, the revised title can make the readers easier to understand the main purpose of the work.
- 2) Many thanks for supporting the changes in Figure 3. In this work, the use of differential charge density as discussed in *J. Chem. Theory Comput.* **2011**, 7, 2498 gave a better representation through this form.
- 3) We are thankful for the constructive comments. The electron paramagnetic resonance under photo-illumination can indeed quantitatively give the corresponding results through the intensity change. This professional suggestion will be considered in our future works.

Dr. Hai-Long Jiang, Professor, FRSC
Department of Chemistry
Hefei National Research Center for Physical Sciences at the Microscale
University of Science and Technology of China (USTC)

Response to Reviewer 2:

General Comment: *“The authors have implemented all suggestions and I believe the manuscript is in good shape for publication. For this reason, I congratulate the authors for the excellent work done.”*

Response: Many thanks for the very positive evaluation and kind encouragement.